# Carbones and Carbon Atom as Ligands in Transition Metal Complexes

**DOI:** 10.3390/molecules25214943

**Published:** 2020-10-26

**Authors:** Lili Zhao, Chaoqun Chai, Wolfgang Petz, Gernot Frenking

**Affiliations:** 1Institute of Advanced Synthesis, School of Chemistry and Molecular Engineering, Jiangsu National Synergetic Innovation Center for Advanced Materials, Nanjing Tech University, Nanjing 211816, China; ias_llzhao@njtech.edu.cn (L.Z.); 201861205094@njtech.edu.cn (C.C.); 2Fachbereich Chemie, Philipps-Universität Marburg, Hans-Meerwein-Strasse 4, D-35043 Marburg, Germany

**Keywords:** carbone complexes, carbido complexes, transition metal complexes, chemical bonding

## Abstract

This review summarizes experimental and theoretical studies of transition metal complexes with two types of novel metal-carbon bonds. One type features complexes with carbones CL_2_ as ligands, where the carbon(0) atom has two electron lone pairs which engage in double (σ and π) donation to the metal atom [M]⇇CL_2_. The second part of this review reports complexes which have a neutral carbon atom C as ligand. Carbido complexes with naked carbon atoms may be considered as endpoint of the series [M]-CR_3_ → [M]-CR_2_ → [M]-CR → [M]-C. This review includes some work on uranium and cerium complexes, but it does not present a complete coverage of actinide and lanthanide complexes with carbone or carbide ligands.

## 1. Introduction

Transition metal compounds with metal-carbon bonds are the backbone of organometallic chemistry. Molecules with M-C single bonds are already known since 1849 when Frankland reported the accidental synthesis of diethyl zinc while attempting to prepare free ethyl radicals [1,2]. Molecules with a [M]=CR_2_ double bond (carbene complexes) or a [M]≡CR triple bond (carbyne complexes) were synthesized much later [3,4,5,6]. Two types of compounds with metal-carbon double or triple bonds having different types of bonds are generally distinguished, which are named after the people who isolated them first. Fischer-type carbene and carbyne complexes are best described in terms of dative bonds following the Dewar–Chatt–Duncan (DCD) model [7,8] [M]⇄CR_2_ and [M^(─)^] 
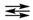
 CR^(+)^, whereas Schrock-type alkylidenes and alkylidynes are assumed to have electron-sharing double and triple bonds [M]=CR_2_ and [M]≡CR [9,10,11].

This review deals with transition metal complexes with metal-carbon bonds to two types of ligands, which have only recently been isolated and theoretically studied. One type of ligand are carbones CL_2_ [12], which are carbon(0) compounds with two dative bonds to a carbon atom in the excited ^1^D state L→$\;\overset{‾}{\underset{¯}{C}}\;$←L where the carbon atom retains its four valence electrons as two lone pairs that can serve as four-electron donors [13,14]. Thus, carbones CL_2_ are four-electron donor ligands whereas carbenes CR_2_ are two-electron donors. Carbenes have a formally [15] vacant p(π) orbital that can accept electrons in donor-acceptor complexes M⇄CR_2_ whereas carbones are double (σ and π) donors in complexes [M]⇇CL_2_. A good Lewis acid acceptor fragment A for a carbene complex has a vacant σ orbital and an occupied π orbital whereas a suitable acceptor for a carbone is a double Lewis acid with vacant σ and π orbitals as shown in Figure 1a,b. If the Lewis acid A has an occupied π orbital, it would lead to π repulsion with the π lone pair of the carbone CL_2_, whereby the repulsive interaction is reduced if L is a good π acceptor (Figure 1c). The two electron lone pairs of a carbone may bind to one or two monodentate Lewis acids A or protons or to a single bidentate Lewis acid as shown in Figure 1. The large second proton affinity is a characteristic feature of carbones, which distinguishes them from carbenes [16]. Examples of all cases are known and are described below.

It is important to realize that the two electron lone-pairs of a carbone CL_2_ may additionally engage in π-backdonation to the ligands L whose strength depends on the availability of vacant π orbitals of the ligands L. Stronger π acceptor ligands L enhance the π-backdonation L←$\;\overset{‾}{\underset{¯}{C}}\;$→L which leads to wider bending angles at the carbon atom (Figure 2). The significant bending of free C(CO)_2_ [17,18] can straightforwardly be explained in terms of dative bonding in carbon suboxide C_3_O_2_ [19,20]. The π-acceptor strength of ligands L thus modulates the donor interaction of the carbone CL_2_.

The following list gives some essential features of carbones and their differences to carbenes. At the same time we want to stress that the distinction between carbenes and carbones are just a useful classification of compounds, which are a helpful model to explain the structures and reactivity of molecules. Nature does not exhibit a strict distinction line and there are complexes with electronic structures that have intermediate features between both classes of compounds. Carbenes and carbones are two ordering principles like ionic and covalent bonding. Intermediate cases are common and yet, the two concepts are essential ingredients of chemistry. The first part of this review summarizes experimental and theoretical work about transition metal complexes with carbone ligands [M]-CL_2_.

Carbones are neutral carbon(0) compounds of the general formula CL_2_, which possess two electron lone pairs of electrons of σ and π symmetry, respectively.Carbones CL_2_ have dative σ bonds L→$\;\overset{‾}{\underset{¯}{C}}$
←L and weaker π backdonation L←$\;\overset{‾}{\underset{¯}{C}}$
→L which resemble donor-acceptor bonds in transition metal complexes.The carbon atom of carbones has very large electron densities and thus, unusually large negative partial charges.In contrast to carbenes, carbones exhibit high first and second proton affinities (PAs) in the region of about 290 and 150–190 kcal/mol, respectively. The second PA is a sensitive probe for the divalent C(0) character of a CL_2_ molecule. Carbones can take up one and two protons with formation of [HCL_2_]^+^ cations or [H_2_CL_2_]^2+^ dications, respectively.Carbones have a bent equilibrium geometry where the bending angle becomes wider when the ligand L is a better π acceptor.Carbones can take up one or two monodentate Lewis acids A building the complexes A←C(L_2_) and A←C(L_2_)→A or one bidentate Lewis acid A⇇C(L_2_).

To the thematic of carbones several review articles were reported previously; A general overview on species that bear two lone pairs of electrons at the same C-center are summarized in [21], transition metal adducts of carbones are described in [22], and those of main group fragments in [23]. Two contributions, [24] and [25], in the series Structure and Bonding (Springer Edition) also deal with carbone transition metal addition compounds.

The second type of transition metal complexes with a carbon ligand features species with a naked neutral carbon atom as a ligand [M]-C, which can be considered as endpoint of the series [M]-CR_3_ → [M]-CR_2_ → [M]-CR → [M]-C. Complexes with negatively charged carbon ligands [M]-C^─^, which are isoelectronic to nitride complexes [M]-N and are termed as carbides, were synthesized in 1997 by Cummins [26]. The first neutral carbon complex [M]-C, which was prepared and structurally characterized was reported in 2002 by by Heppert and co-workers [27]. They isolated the diamagnetic 16 valence electron ruthenium complexes [(PCy_3_)LCl_2_Ru(C)] (L = PCy and 1,3-dimesityl-4,5-dihydroimidazol-2-ylidene; Cy = Cyclohexyl) by a metathesis facilitated reaction. Quantum chemical calculations of model compounds suggested that the Ru-C bond in the complexes is best described by an electron-sharing double bond like in Schrock carbenes, which is reinforced by a donor bond [Ru]$\vec{=}$C**|** [28]. The field of neutral carbon complexes was systematically explored in recent years by Bendix [29]. This review summarizes in its second part the research in transition metal complexes with a naked carbon atom as ligand [M]-C that has been accomplished since 2002. The review includes some work on uranium and cerium complexes, but it does not present a complete coverage of actinide and lanthanide complexes with carbone or carbide ligands.

## 2. Transition Metal Complexes with Carbone Ligands [M]-CL_2_

### 2.1. Transition Metal Addition Compounds of Symmetrical Carbones C(PR_3_)_2_

Among the existing carbones with a symmetric P-C-P skeleton, five species (**1a**–**1e**) are known today as donor ligands to various transition metal fragments as outlined in Figure 3. From other linear or bent carbones with this skeleton, no transition metal complexes are described so far.

In 1961, **1a** was detected by Ramirez [30], and 1b–1d stem from the laboratory of Schmidbaurs group [31]. Later on, a series of related carbones were synthesized, but for which transition metal complexes are unknown so far. Quite recently the new amino substituted carbone **1e** was published together with Zn and Rh addition compounds (See Scheme 1) [32]. In the 31P NMR spectra singlets at about −4.50 (**1a**), −6.70 (**1b**), −29.6 (**1c**), −22.45 (**1d**), and 12.5 ppm (**1e**) confirm the symmetric array of the compounds. All carbones have a bent structure but a linear form of **1a** is realized if crystallized from benzene [33,34]. **1a** has a short P-C distance of 1.633(4) Å and the P-C-P angle amounts to 130.1(6)° [35]. The carbone **1b** exhibits a slightly longer P-C distance of 1.648(4) Å and the introduction of two less bulky methyl groups allows a more acute P-C-P angle of 121.8(3)° [36]. **1d** has similar P-C bond distances of 1.645(12) Å 1.653(14) Å and the acutest P-C-P angle in this series of 116.7(7)° [37,38]. For **1c**, gas phase electron diffraction studies result in a P-C distance of 1.594(3) Å and a P-C-P angle of 147.6(5)° assuming an apparent non-linearity but linearity in the average structure [37]. All structural parameters of **1e** are close to those of **1a** (P-C = 1.632(2) Å, P-C-P angle = 136.5(3)° [32].

In Table 1, transition metal addition compounds between carbones with the P-C-P core are collected. All compounds show longer P-C bonds than the basic carbones as consequence of the competition of the occupied p orbital at C(0) between the two P-σ* orbitals and those of A.

Occupied d orbitals of Ni in the 1a-Ni(CO)_3_ complex elongate the C-Ni bond to a carbone (2.110 Å) [39] but this leads to a relative short bond length to a NHC (1.971 Å) moiety [57]. In contrast, UCl_4_ leads to a short bond to a carbone (2.411 Å) [51] indicating an appreciable U-C double bond character and a long one to a NHC base (2.612 Å) [58,59].

The cation [**1a**-ReO_3_]^+^ holds the longest one with 1.771(8) Å indicating an appreciable C=Re double bond character. This feature applies also in part to **1a**-UCl_4_ and **1c**-W(CO)_2_N_3_ with elongated P-C bonds(See Scheme 2); a partial C-U double bond is confirmed by theoretical calculations. Similar long P-C bonds are found in the trication [**1aH**-Ag-**1aH**]^3+^, in **1a**-(AuCl)_2_(See Scheme 3), and in **1b**-Ni_2_(CO)_5_(See Scheme 4), where the carbone provides each two electrons to two accepting Lewis acids as depicted in Figure 1d.

The P-C-P angles are in the range between 115° and 132° reflecting the required space of the appropriate Lewis acid. The ^31^P NMR shift of the carbone **1a** amounts to about −5 ppm and those of the related addition compounds are shifted to lower fields and range between 4 ppm and 30 ppm. All iron(II) complexes of **1a** and **1b** are paramagnetic and ^31^P NMR spectra could not be obtained.

For the ^31^P NMR spectrum of the carbone **1b**, a shift of −6.70 ppm was recorded [31]. With exception of **1b**-Ni(CO)_3_ which resonate at 2.6 ppm, low field shifts between 12 and 22 ppm were found when **1b** act as a four electron donor [40].

Further, **1e**-ZnCl_2_ (See Scheme 5) [32] and **1a**-ZnI_2_ [53] have closely related structural parameters but exhibit shorter C-Zn bond lengths than to related NHC-addition compounds (Δ = 0.051 Å) [60]. In both compounds a nearly perpendicular array of the ZnX_2_ and the PCP plane are found. No tendency for an additional N-coordination to the amino ligand of **1e** is recorded for the ZnCl_2_ addition compound. In contrast the Rh-C distances in **1e**-Rh(CO)_2_(acac) are longer (Δ = 0.117 Å) than in the corresponding NHC compound [61] and a partial π interaction was found by DFT calculation. Rh also shows no tendency for coordination of the adjacent amino groups [32]. 

### 2.2. Transition Metal Addition Compounds of Carbones C(PR_3_)_2_ with an Additional Pincer Function 

Starting material for **2a** is not the free carbone Ph_2_P-CH_2_-PPh_2_-C-PPh_2_-CH_2_-PPh_2_, which could not be prepared so far, but the dication [Ph_2_P-CH_2_-PPh_2_-CH_2_-PPh_2_-CH_2_-PPh_2_]^2+^ as reported by Peringer [62]. Later on, Sundermeyer studied the deprotonation of the cation [Ph_2_P-CH_2_-PPh_2_-CH-PPh_2_-CH_2_-PPh_2_]^+^ by quantum chemical methods giving more or less stable tautomers of **2a**, see Figure 4. Deprotonation of the tautomer C of **2a** generates the anionic pincer ligand [Ph_2_P-CH-PPh_2_-CH-PPh_2_-CH-PPh_2_]^−^ [**2c]^−^** [63]. The same working group also published the X-ray structure of the pincer ligand **2b** with the P-C-P angle of 133.76(13)° and P-C distances of 1.633(2) and 1.642(2) Å; the ^31^P NMR shift δ = −5.6 ppm [64].

Various cationic complexes where reported with the pincer ligand **2a** (See Figure 4) and group 10 metal halides and one dication with the group 11 metal Au. The ^31^P NMR shifts range between 32 and 41 ppm(See Table 2). As with **1a** the carbone carbon atom of **2a** is basic enough to accept a proton to generate complexes of the type **2aH**-MCl dications with all group 10 elements (See Scheme 6).

A series of complexes with the N,C,N pincer ligand *sym*-bis(2-pyridyl)tetraphenylcarbodiphosphorane (**2b**) were reported recently by the group of Sundermeyer. Remarkable is the molybdenum complex **2b**-[Mo_2_(CO)_7_] in which **2b** provides four pairs of electrons for donation to a Mo_2_ unit with an Mo-Mo separation of 3.0456(5) Å [64]. This coordination mode is continued in a series of dicopper complexes presented by the same working group and prepared as depicted in Scheme 7. The addition of [Cu]PF_6_ to **2b** followed by treatment with two eq. of PR_3_ generated the cationic complexes [**2b**-(CuPPh_3_)](PF_6_)_2_ and [**2b**-(CuP{C_6_H_4_OMe}_3_](PF_6_)_2_, respectively; **2b**-(CuCarb)_2_ was obtained from **2b**-(CuCl)_2_ and two eq. of CarbH/NaO*^t^*Bu (CarbH = carbazol) [63]. 

For the cationic complexes [**2b**-Cu_2_(P-P)]^2+^ the chelating ligands are: DPEPhos = bis[(2-diphenylphosphino)phenyl] ether, XantPhos = 4,5-bis(diphenylphosphino)-9,9-dimethylxanthene, dppf = 1,10-bis(diphenyl-phosphino)ferrocene. The germinal nature of both Cu(I) centers leads to Cu-Cu distances in the range of 2.55–2.67 Å. Most of the Cu(I) complexes show photoluminescence upon irradiation with UV light at room temperature [63].

Further, **2cH**-CuPPh_3_ is an example of a complex with a deprotonated form of **2a** and longer P-C distances are observed due to the protonation of the central carbon atom [63].

### 2.3. Transition Metal Addition Compounds of Carbones C(PR_3_)_2_ with an Additional Ortho Metallated Pincer Function 

The source for the Rh complex **3a**-Rh(PMe_3_)_2_H was the half pincer compound **5a**-Rh(C_6_H_8_) (vide infra) upon reacting with PMe_3_ under loss of cod (see Scheme 8). **3a**-Pt(SMe_2_) forms upon reacting **1a** with [Me_2_Pt(SMe_2_)]_2_ and loss of 4 molecules of CH_4_ [69]. PEt_3_ replaces the labile bonded SMe_2_ group of **3a**-Pt(SMe_2_) to produce **3a**-PtEt_3_, which is transformed with P(OPh)_3_ into **3a**-Pt(OPh)_3_. The dication [**3a**-PtPEt_3_(μ-Ag_2_)Et_3_PPt-**3a**]^2+^ was obtained upon addition of AgOTf to **3a**-PtPEt_3_. According to the carbone C atom as four electron donor the Pt complexes with μ-Ag functions show long Pt-C distances between 1.737 and 1.749 Å (mean values) and the ^31^PNMR shifts are in the narrow range of 33 and 36 ppm (See Table 3) [70]. More complicated is the formation of **3a**-Pt(CO), which stems from the hydrolysis of the related **3a**-Pt(CCl_2_) complex (not isolated) [71].

The carbone complex **3b**-Pt(CO) was obtained from reacting the yldiide platinum complex (see Scheme 9) with 1 atm CO that inserts into the N-Si bond of the yldiide.

### 2.4. Transition Metal Complexes with P-C-P Five Membered Ring 

The carbone **4** (see Figure 6) was obtained by deprotonation of the cation [**4H]^+^**. According to two P atoms in different chemical environments two doublets in the ^31^P NMR spectrum were recorded at δ = 60.0 and 71.5 ppm; ^2^J_PP_ = 153 Hz. From X-ray determination stem the P-C(1) and P-C(2) distances of 1.644(19) and 1.657(17) Å, respectively, and the P-C-P angle amounts to 104.82(10)° [73]. The bond lengths (see Table 4) are close to that reported for the carbone **1a**.

From the cyclic and asymmetric carbone **4** six transition metal complexes (see Scheme 10) are known in which the ligand acts as two electron donor via the C atom. As in the starting compound **4** the P^2^-C bond distances are slightly longer than P^1^-C bond. Addition of CuCl and AuCl(SMe_2_) to **4H**^+^/*t*BuOK generates the compounds **4**-CuO*t*Bu and **4**-AuO*t*Bu, respectively. In CH_3_Cl_2_ or CHCl_3_
**4**-CuO*t*Bu is converted into **4**-CuCl [74]. **4**-Rh(CO)_2_Cl stems from the reaction of **4** with [{RhCl(CO)_2_}_2_] [73]. **4**-CuO*t*Bu and **4**-AuO*t*Bu catalyze the hydroamination or hydroalkoxylation of acrylonitrile [74].

### 2.5. Transition Metal Complexes with Asymmetric P-C-P Ligands 

Several asymmetric carbones with orthometallation (**5a-M**, **5d-M**), with an additional donor function (**5c**), or with a functionalized phenyl ring (**5b**) were reported that form TM complexes (see Figure 7).

The neutral asymmetric carbone **5b** (X = PPh_2_) has the structural parameters P^1^-C = 1.642(2), P^2^-C = 1.636(1) Å, and a P-C-P angle of 140.74(8)° (see Table 5); the P atoms resonate at δ = −6.9 and −3.4 ppm (^2^J_PP_ = 93 Hz) [75]. Those of **5c** are P^1^-C = 1.6416(16) Å, P^2^-C = 1.6398(17) Å, and P-C-P = 133.25(10)° [76]. Three complexes in which the carbone **1a** is half-side orthometallated forming **5a-M** complexes are described [69,73,77].

As depicted in Scheme 11, three neutral complexes of **1a** are known in which one of its phenyl group is orthometallated to produce the **5a-M** core. The ^31^P NMR shift of the unchanged PPh_3_ group range between about 6 and 13 ppm whereas for the orthometallated side shifts between 15 and 40 ppm where recorded. Both P-C distances do not differ markedly and amount to about 1.700 Å.

All complexes shown in Scheme 12 have a further PPh_2_ function at the ortho position of one phenyl group of **1a**. In the complex **5b**-(AuCl)_2_ the carbone provides four electrons for donation with typical long P-C distances of about 1.770 Å [75].

The paramagnetic **5c**-UCl_4_ exhibits a short C-U distance indicative for a double dative bond of the carbone C atom as in **2b**-UCl_4_ and was obtained by reacting UCl_4_ with the dication **5c**-H_2_/NaHMDS. Upon further coordination of the pyridyl group (U-N = 2.537(4) Å) the U atom attains the coordination number 6 [41].

[**5c**-AuPPh_3_]^+^ was obtained from reacting the carbone **5c** with [PPh_3_AuCl]/Na[SbCl_6_] (see Scheme 13). In the cationic complex [**5c**-(CuCl)((AuPPh_3_)]SbF_6_, the carbone **5c** acts as a six-electron donor with a Cu-N distance of 2.267(6) Å and Cu-Au separation of 2.8483(10) Å. The Cu and Cl atoms are each disordered over two positions with occupancy of about 0.8 to 0.2. If CuCl is replaced by AuCl as in [**5c**-(AuCl)(AuPPh_3_)]SbF_6_ the C-AuPPh_3_ distance is slightly elongated and no coordination of the pyridyl N atom is observed. The Au-Au separation is with 3.1274(6) Å too long for a metallophilic interaction. In both compounds, the carbone C atom constitutes a chiral center according to four chemical different substituents and acts as a four-electron donor. The PPh_3_ group resonates between 15 and 27 ppm [76]. In the related symmetric pyridyl-free complex **1a**-(AuCl)_2_, slightly shorter C-Au (2.076(3) Å) were recorded accompanied by longer P-C (1.776(3) Å) bond lengths [51].

### 2.6. Transition Metal Complexes of Carbones with Cyclobutadiene 

The carbones **6a** and **6b** (see Figure 8) can also be seen as an all-carbon four-membered ring bent allene (CBA); **6a** is stable for several hours at −20° but decomposes when warmed up to −5°. The optimized geometry reveals a very acute allene bond angle of 85.0° and coplanarity of the ring carbon atoms including the two nitrogen atoms. The C=C bonds of the allene fragment amount to 1.423 Å and are significantly longer than in typical linear allenes (1.31 Å). Short CN bonds of 1.36 Å indicate some double bond character. The C**C**C carbon atom resonates in the ^13^C NMR spectrum at 151 ppm. The first and second proton affinities (PAs) are very high amounting to 307 and 152 kcal/mol [79]. 

The molecular orbitals show that the HOMO and HOMO-1 have clearly the largest coefficients at the central carbon atom and exhibit the typical shape of lone-pair molecular orbitals with σ (HOMO) and π (HOMO-1) symmetry; however, with reversed order with respect to CDPs and CDCs. To emphasize the proximity of **6** to CDP carbones, we use the same symbolism mimicking a metal.

The free CBA **6b** could not be obtained, but only the cationic **6bH^+^** and **6bH_2_^2+^** are known and used as starting compounds for the syntheses of the related transition metal complexes [80].

The ^13^C NMR shifts of the central carbon atom are shifted to higher fields relative to the starting free carbone ranging between 124 and 139 ppm (see Table 6).

All complexes of the CBA **6a** where obtained by reacting the freshly prepared free carbone **6a** at −20° with [{MCl(cod)}_2_] complexes (M = Rh, Ir). The cod ligand can be replaced by bubbling CO through solutions of **6a**-MCl(cod) to produce the related **6a**-MCl(CO)_2_ compounds (see Scheme 14) [79].

Transition metal complexes with **6b** as ligand were obtained by reacting 1,1,2,4-tetrapiperidino-1-buten-3-yne with (a) [(tht)AuCl], (b) [RhCl(CO)_2_]_2_, and (c) [(NMe_3_)W(CO)_5_] during the reaction rearrangement of the starting buten-3-yne to **6b** has occurred [80].

### 2.7. Carbodicyclopropenylidene

Stephan described the first carbodicarbene stabilized by flanking cyclopropylidenes, named carbodicyclopropylidene **7** (see Figure 9) [81].

Neither the neutral singlet 1,2-diphenylcyclopropenylidene as carbene ligand L in **7** nor the carbone tetraphenylcarbodicyclopropenyliden (CDC) **7** itself are stable compounds at room temperature. The free carbene L has only been observed in an argon matrix isolated at 10 K and **7** could be characterized in solution by low temperature NMR spectroscopy; for the central carbon atom a ^13^C NMR shift at δ = 133 ppm was recorded at −60 °C. 

The first and second proton affinities of **7** were determined to be 283 and 153 kcal/mol, respectively. The molecular structure of **7** was determined by computational methods. Calculations reveal that the central carbon atom is in a linear environment the C-C distances were calculated at 1.308 Å and the C-C-C angle to 180°. The energy difference between the linear allenic structure and the bent arrangement is shallow amounting to 6.6 kcal/mol for a bending angle of 140° and 10 kcal/mol for 130°. The highest occupied molecular orbital (HOMO) and HOMO-1 of **7** are degenerate and incorporate the p(π) orbitals of the C2-C1-C2a fragment.

The central C atom is more negatively charged (−0.19 a.u.) than the adjacent C atoms, suggesting nucleophilic character [81].

The addition compounds [**7**-AuNHC-Ad](OTf) and [**7**-AuNHC-Dipp](OTf) (see Table 7) were prepared from reacting [**7H**]^+^ with KHMDS and the related (NHC)AuOTf at −45°(see Scheme 15) [81].

### 2.8. Carbodicarbenes

Carbodicarbenes, CDCs, are neutral compounds where a bare carbon atom with its four electrons is stabilized by two NHC ligands which plays the role of a phosphine group as in carbodiphosphoranes, CDPs. Theoretical studies have demonstrated that this class of compounds could be stable and their existence was predicted by Frenking [82] and short times later realized by the group of Bertrand [83].

Structural and spectroscopic parameters of the following symmetric CDCs (see Figure 10) are available: **8a,** C-C = 1.343(2) Å, C-C-C = 134.8(2)°, ^13^C NMR 110.2 ppm [83]; **8b,** C-C = 1.333(2) Å and 1.324(2) Å, C-C-C = 143.61(15)° [84]; **8c,** C-C = 1.335(5) Å, C-C-C = 136.6(5)° (see Table 8) [85].

Structural parameters of the unsymmetrical CDCs (see Figure 11) are: **8e**, C-C = 1.3401(16) Å and 1.3455(16), C-C-C 137.55(12)°. For **8f**, no data are available [90]. **8g**: C-C = 1.344(3) Å and, 1.318(3) Å, C-C-C = 146.11(19)° [90]. **8h** was obtained at −60° by reacting **8hH^+^** with KMDS, and characterized spectroscopically. On warming to room temperature, it dimerizes. ^13^C NMR: δ = 105.5 ppm (see Table 8) [91].

Further, **8a**-RhCl(CO)_2_ was prepared by addition of a suspension of **8a** (see Scheme 16) in benzene to a solution of [RhCl(CO)_2_]_2_ [83]. [**8b**-Fe_0.5_]^2+^ contains Fe^2+^ in octahedral environment coordinated by two molecules of **8b**. Fe(II) can be successively oxidized to the corresponding tri-, tetra-, and pentacationic species [87].

The addition compounds **8c**-RhCl(CO)_2_ and **8d**-RhCl(CO)_2_ where obtained upon reacting the appropriate carbone **8c** or **8d** with [RhCl(CO)_2_]_2_. Similarly, the addition of [Pd(allyl)Cl]_2_ to **8c** leads to the allyl complex **8c**-PdCl(C_3_H_5_) [85]. 

As depicted in Scheme 17, introduction of PdCl_2_P(OiPr)_3_ to **8e** afforded the complex **8e**-PdCl_2_P(OiPr)_3_; it features a square planar Pd center with a short interatomic distance of one phosphite oxygen atom and the carbon atom of the NHC molecule of 2.890 Å that is smaller than the sum of van der Waals radii. This indicates strong attractive interaction between the atoms [88]. The three Pd complexes **8e**-PdCl_2_PPh_3_, **8e**-PdCl_2_PTol_3_, and **8e**-PdCl_2_PCy_3_ were obtained by reacting the carbone **8e** with the appropriate PdCl_2_PR_3_; between the NHC and the aromatic phosphine substituents (Ph or Tol) an unexpected π-π interaction was detected. One Ph and one Tol group are nearly parallel to the imidazole rings with centroid-centroid distances of 3.25 Å (Ph) and 3.30 Å (Tol), respectively [89].

**8f**-RhCl(CO)_2_ and **8g**-RhCl(CO)_2_ stem from reacting the appropriate carbone with [RhCl(CO)_2_]_2_ [90]. The cod ligand of [Ir(cod)Cl]_2_ was replaced by bubbling CO through a mixture with **8h** to generate the complex **8h**-IrCl(CO)_2_ [91]. 

Some experimental findings indicate that carbodicarbenes also have catalytic properties for a wide range of transformations, which are currently being actively studied by several groups. Examples have been reported such as hydrogenation of inert olefins [92], C-C cross-coupling reactions [84], intermolecular hydroamination [93] and hydroheteroarylation [94]. It seems that this area is still in an infant stadium and it can be expected that CDCs may be found useful as catalyst for other reactions.

### 2.9. Tridentate Cyclic Diphosphino CDCs

The carbones **9a** and **9b** in Figure 12 are functionalized carbodicarbene in which the donating carbon atom is part of a seven membered ring.

The neutral **9a** and **9b** could not be isolated, source for transition metal complexes are the related cations **9aH^+^** and **9bH^+^** (see Table 9) [93].

The neutral complexes **9a**-RhCl and **9b**-RhCl (see Scheme 18) where prepared upon reacting the cations **9aH^+^** or **9bH**^+^, respectively with [Rh(cod)Cl]_2_/NaOMe; if treated with AgBF_4_/MeCN the cationic spezies [**9a**-Rh(MeCN)]BF_4_ and [**9b**-Rh(MeCN)]BF_4_, respectively, were isolated. The related carbonyl complexes [**9a**-Rh(CO)]BF_4_ and [**9b**-Rh(CO)]BF_4_ formed similarly upon reaction with [Rh(CO)_2_Cl]_2_/NaOMe [93]. The styrene complex [**9a**-Rh(styrene)]^+^ was obtained upon treating the related chloro complex with styrene/NaBAr_4_; the styrene complex catalyzes the hydroarylation of dienes. Protonation of [**9a**-Rh(CO)]^+^ with HBF_4_·OEt_2_ generates [**9aH**-Rh(CO)]^2+^ in which the carbone acts as four-electron donor [94].

### 2.10. Tetraaminoallene (TAA) Transition Metal Complexes 

The ^13^C NMR shift of the central carbon atom amounts to 142.8 ppm. The first and second PAs of **10** are 282.5 and 151.6 kcal/mol, respectively [16,82].

The salt [**10**-AuPPh_3_]SbF_6_ in Scheme 19 is the only transition metal complex of TAA (see Figure 13), which has been reported so far. Both carbene moieties are planar, but are tilted relative to each other, to relieve allylic strain. The Au-C bond lengths amounts to 2.072(3) Å and the slightly different C-C dative bonds has interatomic distances of 1.406(5) and 1.424(5) Å. The central C-C-C bond angle is reported with 118.5(3)° [95].

### 2.11. Transition Metal Complexes of Carbones with the P-C-C Skeleton 

Mixed carbene-phosphine stabilized carbones from the working group of Bestmann (1974) and Alkarazo (2009).

The crystal structure of **11a** in Figure 14 reveals a planar configuration of the carbene ligand C(OEt)_2_. Short P-C and C-C distances indicate some p back donation; P-C = 1.682(4)Å, C-C = 1.316(10) Å, C-C-C 125.6° (see Table 10) [97].

The neutral Rh complex **11a**-RhCl(CO)_2_ was obtained from reacting the carbone **11a** with [Rh(CO)_2_Cl]_2_. Similarly, the complex **11b**-AuCl results from reaction of **11b** with AuCl(SMe_2_) (Scheme 20) [98].

### 2.12. Transition Metal Complexes of Carbones with the P-C-Si Skeleton 

The neutral compound **12** in Figure 15 is a carbone in which the C(0) atom is stabilized by a donor stabilized silylene and a phosphine ligand.

The crystal structure of a related compound to **12** (a cyclopentene instead of a cyclohexene ring) shows a P-C distance of 1.6226(4) Å and Si-C distance of 1.6844(4) Å; the Si-C-P angle amounts to 140.03(3)°. 

Addition of CuCl generates the complex **12**-CuCl. No spectroscopic or structural details are available [99].

### 2.13. Transition Metal Complexes of Carbones with the P-C-S Skeleton 

A series of carbones (**13a**, **13b**) in Figure 16 based on a P-C-S core containing the neutral S(IV) ligands SPh_2_=NMe (Figure 16) were reported by Fujii [100]. 

Crystal structures and ^31^P NMR shifts of the following basic carbones are available (see Table 11): **13a,** δ = −2.64 ppm; **13b**, δ = −1.39 ppm, P-C = 1.663(2) Å, S-C = 1.602(2) Å, P-C-S = 125.59(15)°. The authors revealed a high electron density at the central carbon atom.

The addition products **13a**-AgCl and **13b**-AgCl were obtained from reacting [**13aH**]^+^ or [**13bH**]^+^, respectively with ion exchange resin (Cl^−^ form) and Ag_2_O/CH_2_Cl_2_. For the other products see Scheme 21 [100].

Addition of TM fragments to **13a** or **13b** in Scheme 21 elongates P-C and S-C bond length as reported for **1a.** That of [**13bH**-AuPPh_3_](OTf)_2_ in which **13b** acts as four-electron donor are elongated to normal single bonds [100].

### 2.14. Transition Metal Complex with a P-C-S Core Possessing a Neutral S(II) Ligand

The carbone **14** in Figure 17 contains a phosphine and a S(II) ligand with a free pair of electrons to stabilize the C(0) atom. However, the bare **14** could not be isolated, but only the protonated cation [**14H**]^+^ and used as starting material [101].

The transition metal complex [**14**-CuN(SiMe_3_)_2_](OTf) was prepared upon reacting [**14H**]+ with KHMDS/CuCl. X-ray analysis reveals a Cu-C distance of 1.903(4) Å and the P-C and S-C distances amount to 1.709(5) and 1.677(5) Å, respectively. As found in carbone addition compounds of **13a** and **13b** the P-C distance is longer than the S-C distance. An acute P-C-S angle of 115.3(2)° was recorded. The ^31^P NMR signal is shifted to lower fields at 66.5 ppm [101].

### 2.15. Transition Metal Complexes of Carbones with the S-C-S Skeleton 

In the carbones **15** (carbodisulfanes, CDS) the central carbon atom is stabilized by two neutral S(II) ligands (**15a**), or S(II), S(IV) groups (**15b**), or two S(IV) (**15c**) ligands (see Figure 18). 

The molecular structure of **15a** was investigated computationally (see Table 12) [102]. For the carbones the following parameters were recorded: **15b**, C-S^II^ 1.707(2), C-S^IV^ 1.648(2), S-C-S 106.67(14). ^13^C NMR, *δ* = 35.4 ppm [103]. **15c**, S-C 1.635(4), 1.636(2); S-C-S 116.8(2) [104]. Similar to CDCs the first and second PAs of **15b** amount to 288.0 and 184.4 kcal/mol, respectively.

**15a**-AgCl was obtained from [**15aH**]^+^ upon treating with Ag_2_O/CH_2_Cl_2_. The salt [**15a**-AuPPh_3_]OTf formed reacting the bare **15a** with AuCl(PPh_3_) followed by addition of NaTfO in THF. [**15a**-(AuPPh_3_)_2_](OTf)_2_ and [**15aH**-AuPPh_3_](SbF_6_) are sketched in Scheme 22 [102]. 

[**15b**-AuPPh_3_]OTf was obtained analogously formed from reacting **15b** with AuCl(PPh_3_) followed by addition of NaTfO in THF. For the other compounds, see Scheme 23 [102].

The preparation of [**15c**-AuPPh_3_]OTf and **15c**-AgCl follows the procedure outlined for the related **15b** compounds [102]. For the other compounds, see Scheme 24 [102,105]. The hetero hexametallic cluster {[**15c**-(AuPPh_3_)_2_AgOTf](OTf)_4_}_2_ is supported by two carbone ligands that adopt a *κ^4^C,C’,N,N’* coordination mode. The Au-Ag separation amounts to 3.003 Å [102].

^13^C NMR signals of the donating C(0) atoms (if available) of all addition compounds of **15a** to **15c** are less shielded than that of the basic carbones [102].

### 2.16. Transition Metal Complexes of Carbones with the S-C-Se Skeleton (16)

Compound **16** in Figure 19 is the first carbone containing a Se(II) compound together with a S(IV) one as ligand for stabilization of a C(0) atom.

The tetranuclear complex [**16**-Ag_4_-**16**]^4+^ contains a rhomboidal [Ag_4_]^4+^ core surrounded by two carbones **16** (see Table 13). In this and in [**16H**-Ag-**16H**]^3+^ the donating C(0) acts as a four-electron donor (see Scheme 25) [105].

## 3. Transition Metal Carbido Complexes [M]-C

The second part of this review summarizes the research of transition metal complexes with a naked carbon atom as ligand [M]-C. They are often termed as carbides, but the bonding situation is clearly different from well-known carbides of the alkaline and alkaline earth elements E, which are salt compounds of acetylene E_n_C_2_. The electron configuration of carbon atom in the ^1^D state (2s^2^2p_x_^2^2p_y_^0^2p_z_^0^) is perfectly suited for dative bonding with a transition metal following the DCD model [7] in terms of σ donation and π backdonation [M] 
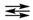
 C**|**. Carbon complexes [M]-C may thus be considered as carbone complexes [M]-CL_2_ without the ligands L at the carbon atoms. A theoretical study showed in 2000 that the 18 valence electron (VE) complex [(CO)_4_Fe(C)] is an energy minimum structure with a rather strong Fe-C bond [106]. However, such 18 VE systems could not be synthesized as isolated species but were only found as ligands where the lone-pair electron at the carbon atom serves as donor (see below). It seems that the electron lone-pair at carbon in the 18 VE complexes [M]-C makes the adducts too reactive to become isolated.

It came as a surprise when Heppert and co-workers reported in 2002 the first neutral adducts with a naked carbon atom as a ligand, which are the formally 16 VE diamagnetic ruthenium complexes [(PCy_3_)LCl_2_Ru(C)] (L= PCy and 1,3-dimesityl-4,5-dihydroimidazol-2-ylidene; Cy = Cyclohexyl) [27]. A subsequent bonding analysis of the model compound [(Me_3_P)_2_Cl_2_Ru-C] considered five different models A–E for the Ru-C bonds that are shown in Figure 20 [28]. It turned out that the best description for the bonding interactions is a combination of electron-sharing and dative bonds. An energy decomposition analysis [107] suggested that the model B provides the most faithful account of the bond, where the σ bond and the π bond in the Cl_2_M plane come from electron-sharing interactions Cl_2_M=C whereas the π bond in the P_2_M plane is due to backdonation (Me_3_P)_2_Ru→C. The compounds [(PCy_3_)LCl_2_Ru(C)] should therefore be considered as 18 VE Ru(IV) adducts. The following section summarizes the research of transition metal complexes with a naked carbon atom as ligand [M]-C that has been accomplished since 2002. 

### 3.1. The System RuCl_2_(PCy_3_)_2_C ([Ru]C)

By far the most known complexes with carbido ligands that have been synthesized and structurally characterized are ruthenium adducts. The progress in the chemistry of ruthenium carbido complexes was reviewed in 2012 by Takemoto and Matsuzaka [108]. In the following, we summarize the present knowledge on ruthenium carbido complexes which has been reported in the literature.

The X-ray analysis of [Ru]C in Figure 21 exhibits a Ru-C distance of 1.632(6) Å. A signal at 471.8 ppm was attributed to the ligand carbon atom [109]. A general route to carbon complexes is described in [110].

Addition of PdCl_2_(SMe_2_)_2_ gives the complex [Ru]C→PdCl_2_(SMe_2_), while with Mo(CO)_5_(NMe_3_) the carbonyl complex [Ru]C→Mo(CO)_5_ is generated (see Table 14) [29,109]. A series of [Ru]C→PtCl_2_L complexes were obtained by Bendix from reacting the dimeric complex {[Ru]C→PtCl_2_]_2_ with various ligands L (L = PPh_3_, PCy_3_, P(OPh)_3_, AsPh_3_, CN^t^Bu, CNCy). Complexes with bridging ligands L such as {[Ru]C→PtCl_2_]_2_bipy, {[Ru]C→PtCl_2_]_2_pyz, and {[Ru]C→PtCl_2_]_2_pym formed upon displacing ethylene from the related (C_2_H_4_)PtCl_2_-L-PtCl_2_(C_2_H_4_) by [Ru]C. {[Ru]C→PtCl]_2_(μ-Cl)pz results from an ethylene complex and [Ru]C as depicted in Scheme 26 [111]. A series of Pt, Pd, Rh, Ir, Ag, Ru complexes were presented by Bendix with X-ray data and ^13^C NMR shifts of the ligand carbon atom ranging between 340 and 412 ppm [112]. Sulfur containing TM complexes with the metals Pd, Pt, Au, and Cu stem from the same laboratory. The sulfur ligands are ttcn = 1,4,7-trithiacyclononane and S_4_(MCp*)_3_ (see Figure 22) [113].

### 3.2. The System RuCl_2_(PCy_3_)(NHC)C (^NHC^[Ru]C)

The X-ray analysis of ^NHC^[Ru]C in Figure 23 exhibits a Ru-C distance of 1.605(2) Å. A signal at 471.5 ppm was attributed to the ligand carbon atom. No addition compounds were described so far [27].

### 3.3. The System (NHC)Cl_3_RuC^−^ (^NHC^[Ru]^−^C)

Treating the carbene complex (NHC)Cl_2_(PCy_3_)Ru=CH_2_ in Figure 24 at 55° in benzene generated the neutral complex depicted in Figure 25. X-ray analysis revealed a Ru^1^-C distance of 1.698(4) Å and the Ru^2^-C distance of 1.875(4) Å with a Ru-C-Ru angle of 160.3(2)°. In the ^13^C NMR the bridging C atom resonates at the typical value of 414.0 ppm [114].

### 3.4. The system RuClX(PCy_3_)_2_C ([Ru]XC)

Various carbido complexes were reported in which one or both chloride ions in [Ru]C are replaced by X (X = Br, I, CN, NCO, NCS) (see Figure 26). {[Ru](MeCN)C}OTf is the first cationic carbido complex which is also starting point for most of the substituted carbido complexes. X-ray data for {[Ru](MeCN)C}OTf, [Ru](CN)_2_C, [Ru](Br)C, and [Ru](NCO)C are available (see Table 15) **[115]**.

### 3.5. The Systems OsCl_2_(PCy_3_)_2_C and OsI_2_(PCy_3_)_2_C ([OsX]C)

The carbido complexes [OsX]C in Figure 27 were studied by X-ray analysis. The most important structural parameter is the Os-C separation, which for X = Cl amounts to 1.689(5) Å [116]. Single-crystal X-ray diffraction reveals that molecular [OsX]C adopts an approximately square-pyramidal core geometry, with the carbido ligand occupying the apical position and a short Os-C bond. In the ^13^C NMR spectrum the signal at 471.8 ppm for X = Cl was attributed to the ligand carbon atom. It was synthesized via S-atom abstraction from the thiocarbonyl complex Os(CS)(PCy_3_)_2_Cl_2_ by Ta(OSi-*t*-Bu_3_)_3_. The diiodo derivative was synthesized from [OsCl]C upon reacting with 10 eq of Me_3_SiI and exhibits a ^13^C NMR signal at 446.14 ppm.

### 3.6. The System [Tp*Mo(CO)_3_≡C]^−^ ([Mo]^−^C)

The reaction between Tp*Mo(CO)_2_CCl (see Figure 28) and KFeCp(CO)_2_ generates the carbido complex [Mo]C→FeCp(CO)_2_ (see Table 16) [117]; see alternative synthesis from Tp*Mo(CO)_2_C-Li and ClFeCp(CO)_2_ [118]. When Tp*Mo(CO)_2_CSe was allowed to react with [Ir(NCMe)(CO)(PPh_3_)_2_]BF_4_ the tetranuclear carbido complex (μ-Se_2_)[Ir_2_-{[Mo]C}_2_(CO)_2_(PPh_3_)_2_] was obtained (see Figure 29) [119]. A solution of Tp*Mo(CO)_2_CBr in THF was treated with BuLi followed by addition of HgCl_2_ resulted in the formation of the carbido complex [Mo]C→Hg←C[Mo] [120]. The platinum complex [Mo]C→Pt(PPh_3_)_2_Br was prepared from reacting [(HB(pz)_3_]Mo(CO)_2_CBr with [(PPh_3_)_2_Pt(C_2_H_4_)] [121].

### 3.7. Unique Mo Carbido Complex 

A further unique carbido complex was described recently as shown in Figure 30. A signal at 360.8 ppm in the ^13^C NMR spectrum was assigned to the ligand carbon atom [123].

### 3.8. The System [Tp*W(CO)_3_≡C]^−^ ([W]^−^C)

Reaction of [W]C-Li(THF) with NiCl_2_(PEt_3_)_2_ produced the complex [W]C→NiCl(PEt_3_)_2_ in Figure 31 [124]. Similarly, with [W]C-Li(THF) and FeCl(CO)_2_Cp or HgCl_2_ the compounds [W]C→Fe(CO)_2_Cp and [W]C→Hg←C[W], respectively, were obtained. [W]C→AuPEt_3_ was prepared from reacting [W]C→SnMe_3_ with AuCl(SMe_2_) followed by addition of PEt_3_. A similar reaction with AuCl(PPh_3_) yielded [W]C→AuPPh_3_. [W]C→AuAsPh_3_ and [W]C→AuPPh_3_ form a tetrameric assembly as depicted in Figure 32. The X-ray analysis of the tetrameric unit revealed Au-C distances of 1.995 and 2.078 Å and the W-C distance is 1.877 Å [122].

The terpyridine complex salt {[W]C→Pt(terpy)}PF_6_ was obtained from [W]C-Li and [PtCl(terpy)]PF_6_; the neutral complex [W]C→PtCl(terpyC[W]) (see Figure 33) was prepared from the same starting material and [PtCl_2_(phen)] (see Table 17) [125].

### 3.9. The Systems N_3_MoC and O_3_MoC

The potassium salt of ^N^MOC^−^ in Figure 34 is dimeric with two K^+^ ions bridging two anions and can be transformed with the crown ethers 2.0-benzo-15-crown-5 and 1.0 2,2,2-crypt into the related ion pairs. X-ray analysis of the crown ether salt revealed a Mo-C distance of 1.713(9) Å [26,126].

The complex [^O^W]C→Ru(CO)_2_Cp was prepared from reacting [^O^W]C-Et with Ru(C≡CMe)(CO)_2_Cp under loss of MeCCEt. The ligand C atom resonates at 237.3 ppm (^1^*J*_WC_ = 290.1 Hz). Distances are W-C = 1.75(2) Å, Ru-C = 2.09(2) Å and the W-C-Ru angle amounts to 177(2) ° [127].

### 3.10. Symmetrically Bridged Carbido Complexes M=C=M

#### 3.10.1. The Fe=C=Fe Core

[Fe(TPP)]_2_C was obtained from Fe^III^(TPP)Cl in the presence of iron powder by reacting with CI_4_ (TPP = 5, 10, 15, 20-tetraphenylporphyrin; according to Fe^II^ the complex is diamagnetic [128]. The complex was also obtained upon reacting Fe(TPP) with Me_3_SiCCl_3_ [129]; see also [130]. An X-ray analysis was performed in [131] and later in [130]. The Mössbauer spectrum is published in [132]. [Fe(TTP)]_2_C (TTP = tetratolylporphyrine) was similarly obtained from Fe(TTP) with Me_3_SiCCl_3_ [129]. [Fe(oep)]_2_C (oep = octaethylporphyrine) was prepared from [ClFe(oep)] and HCCl_3_ and studied by X-ray analysis ans Mössbauer spectroscopy (see Table 18) [132].

The mixed carbido compounds (TPP)Fe=C=Fe(CO)_4_, and (TCNP)Fe=C=Fe(CO)_4_ (TCNP = Tetrakis-p-cyanophenylporphyrinate) were synthesized from [(TPP)FeCCl_2_] or (TCNP)FeCCl_2_ and [Na_2_Fe(CO)_4_]; characterization proceeded via IR spectroscopy [121].

[Fe(pc)]_2_C was prepared from [ClFe(pc)]^−^ and KOH/HCCl_3_ [132], or from Fe(pc) and CI_4_ in the presence of sodium dithionite [95,136], see also [134]. It also forms upon hydrolysis of (Bu_4_N)_2_{[(F)Fe(pc)]_2_C}in acetone [135]. Oxidation with I_2_ generates {[Fe(pc)]_2_C}(I_3_)_0.66_ which was characterized by IR, Mössbauer spectroscopy and powder X-ray diffraction [95]. 

A series of six-coordinate N-Base adducts of μ-carbido phthalocyanine complexes were reported. The pyridine adduct [(py)Fe(pc)]_2_C was obtained y dissolution of [Fe(pc)]_2_C in warm pyridine [133] and characterized by Mössbauer spectroscopy [136] and X-ray analysis [133]. [Fe(pc)(1-meim)}_2_C was similarly obtained as the TPP derivate; starting with pcFe and CI_4_ followed by addition of sodium dithionite gave the μ-carbido bridged dimer; an X-ray diffraction analysis was reportedd (1-meim = 1-methylimidazole, pc = phthalocyanine) [134]. [(4-Mepy)Fe(pc)]_2_C and [(pip)Fe(pc)]_2_C were similarly obtained and studied by IR and Mössbauer spectroscopy [136].

[(thf)Fe(pc)]_2_C forms on dissolving [Fe(pc)] in THF. The asymmetric μ-carbido complex [(thf)(TPP)Fe=C=Fe(pc)(thf)] stems from the reaction of [FeCCl_2_(TPP)] with [Fe(pc)]-; both compounds were characterized by X-ray analyses [130].

Anionic six-coordinate μ-carbido complexes (Bu_4_N)_2_{[(hal)Fe(pc)]_2_C}were reported (hal = F. Cl. Br) and obtained from reacting [Fe(pc)]_2_C with (Bu_4_N)(hal) (F: RT, Cl: 115°, Br: 140°) in solution (F) and in a melt [135].

#### 3.10.2. The Rh=C=Rh Core

[Rh(PEt_3_)_2_(SGePh_3_)]_2_C was obtained upon reacting Rh(PEt_3_)_2_(SGePh_3_)CS with Rh(PEt_3_)_3_(Bpin) via the intermediate mixed carbido complex (SGePh_3_)(PEt_3_)_2_Rh=C=Rh(PEt_3_)_2_(SBpin) which rearranges to this complex and [Rh(PEt_3_)_2_(SBpin)]_2_. The X-ray analysis was performed (see Table 19) [137] [Rh(PEt_3_)_2_(SBpin)]_2_C was prepared earlier by the same working group from Rh(PEt_3_)_3_(Bpin) and 0,5 eq of CS_2_ (X-ray data (see Table 19). Addition of MeOH generated the carbido complex [Rh(PEt_3_)_2_(SH)]_2_C [138]. [Rh(Cl)(PPh_3_)_2_]_2_C resulted from reacting the thiocarbonyl complex Rh(Cl)(PPh_3_)_2_CS with HBCat. The central C atom resonates at 424 ppm (t, ^1^J_RhC_ = 47 Hz). In the chloro complex the chloride ion can be replaced with K[(H_2_B(pz)_2_], K[(H_2_B(pzMe_2_)_2_], or K[(HB(pz)_3_] to produce the carbido complexes [Rh(H_2_B(pz)_2_)(PPh_3_)]_2_C, [Rh(H_2_B(pzMe_2_)_2_)(PPh_3_)]_2_C, and [Rh(HB(pz)_3_)(PPh_3_)]_2_C, respectively (see Figure 35). The unusual asymmetric carbido complex [Rh_2_H(μ-C)(μ-C_6_H_4_PPh_2_-2){HB(pzMe_2_)_3_}_2_] contains a Rh^I^ atom with a shorter Rh-C distance, while the Rh^III^ –C distance is longer [139].

#### 3.10.3. The Ru=C=Ru Core

The tetranuclear carbido complex [Ru(PEt_3_)Cl(μ-Cl_3_)RuAr]_2_C was prepared from the reaction of [(*p*-cymene)Ru(μ-Cl)_3_RuCl(C_2_H_4_)-(PCy_3_)] with HCCH in THF. X-ray analysis adopts Ru-C distances of 1.877(9) Å and a Ru-C-Ru angle of 178.8(9)°(see Figure 36) [140]. 

Five coordinate [Ru(pc)]_2_C with pc = phthalocyaninate was obtained from H[RuCl_2_(pc)] and CCl_2_ (in situ from KOH/HCCl_3_) [132]. The related pyridine adduct with six-coordinate Ru(IV) [(py)Ru(pc)]_2_C was obtained upon dissolution of [Ru(pc)]_2_C in warm pyridine. X-ray analysis revealed a Ru-C distance of 1.77(1) Å and a Ru-C-Ru angle of 174.5(8)° [136].

#### 3.10.4. The Re=C=Re Core

The unique carbido complex [Re(CO)_2_Cp]_2_C in Figure 37 results from reaction of [Re(thf)(CO)_2_(η-C_5_H_5_)], CS_2_, and PPh_3_ (with the aim of the thiocarbonyl complex [Re(CS)(CO)(η-C_5_H_5_)]) as by-product in small amounts. X-ray analysis revealed Re-C distances of 1.882(14) and 1.881(14) Å and a Re-C-Re angle of 173.3(7)°. A ^13^C NMR shift for the bridging carbon atom at *δ* = 436.4 ppm was measured [141].

#### 3.10.5. The W=C=W Core

The oxo complex (*^t^*Bu_3_SiO)_2_(O)W=C=WCl_2_(OSi*^t^*Bu_3_)_2_ in Figure 38 formed in high yield from thermolysis of [(siloxo)_2_Cl(CO)W]_2_ in toluene with loss of CO; in the ^13^C NMR spectrum the carbide C atom resonates at *δ* = 379.14 ppm (J_WC_ = 200, 180 Hz). Degradation of the (silox)_4_C1_2_W_2_(CNAr) complex afforded the imido μ-carbido compound (*^t^*Bu_3_SiO)_2_(NR)W=C=WCl_2_(OSi*^t^*Bu_3_)_2_; the ^13^C NMR shift of the μ-C atom appears at *δ* = 406.25 ppm. X-ray analysis revealed a tetrahedral tungsten core with a W-C distance of 1.994(17) Å (W_1_) and a distorted square-pyramidal tungsten core with a shorter distance of 1.796(17) Å (W_2_). The W-C-W bond angle amounts to 176.0(12)° [142].

### 3.11. Asymmetrically Bridged Carbido Complex Fe=C=M

#### 3.11.1. The Fe=C=Re Core

The asymmetrical carbido complex (TPP)Fe=C=Re(CO)_4_Re(CO)_5_ in Figure 39 was prepared upon reacting the dichlorocarbene complex (TPP)Fe=CCl_2_ with 2 eq of pentacarbonylrhenate, [Re(CO)_5_]^−^, under release of CO and 2 Cl^−^; TPP is tetraphenylporphyrin. Crystals were analyzed by X-ray diffraction and revealed a Fe=C distance of 1.605(13) Å and a C=Re distance of 1.957(12) Å. The Fe-C-Re angle amounts to 173.3(9)°; the Fe-C distance is somewhat smaller than in [(TPP)Fe]_2_C and the Re-C distance is appreciable longer than in [Re(CO)_2_Cp]_2_C. In the ^13^C NMR spectrum the central carbido C atom resonates at 211.7 ppm [143].

#### 3.11.2. The Fe=C=Mn Core

The carbido bridged di-manganese complex (TCNP)Fe=C=Mn_2_(CO)_9_ (TCNP = tetrakis(p-cyanophenyl)porphyrinate) (see Figure 40) was synthesized from [(TCNP)Fe=CCl_2_] and two eq. of Na(Mn(CO)_5_ in THF and characterized with elemental analysis, IR, and UV spectroscopy [121].

#### 3.11.3. The Fe=C=Cr Core

Two compounds with the Fe=C=Cr core have been reported by the group of Beck and characterized by elemental analysis, IR, and UV spectroscopy. Thus, (TPP)Fe=C=Cr(CO)_5_ and (TAP)Fe=C=Cr(CO)_5_ (see Figure 41) were prepared upon reacting the related dichlorocarbene iron complexes [(L)Fe=CCl_2_] with Na_2_[Cr(CO)_5_] in THF (TAP = tetrakis(p-methoxyphenyl)porphyrinate) [121].

## 4. Conclusions

The experimental and theoretical research with regard transition metal complexes with carbone ligands [M]-CL_2_ and carbido complexes [M]-C has blossomed in the recent past and it can be foreseen that it will remain a very active area of organometallic chemistry in the future. The well-known family of transition metal complexes with C1-bonded carbon ligands that comprise alkyl (CR_3_), carbene (CR_2_), and carbyne (CR) groups has been extended by carbones (CL_2_) and carbido (C) ligands. The summary of recent work, which is described in this review, indicates that carbone and carbido complexes are still largely terra incognita and that many new discoveries can be expected.

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
