# Peer review of "Carbones and Carbon Atom as Ligands in Transition Metal Complexes"

_molecules, 2020, doi:10.3390/molecules25214943_

Round 1

Reviewer 1 Report

I really liked the tutorial nature of this review from which I have learnt many new things and also refreshed my memory on concepts that we are all aware of since the birth of organometallic chemistry. I do not see any serious issues with the introduction of this review article. The organization is nice, the selection of the appropriate schemes/figures is correct and the flow of the readability in whole is in good balance.

The compound 1a-UCl4 (and several other in the ms) is not a transition metal complex, therefore it should be literally removed from the paper. However, this example is actually the reason for kindly asking the authors to consider the following points:

- Are there any carbones and carbon atom as ligands in lanthanide and/or actinide chemistry? If yes, my opinion is that the authors need to reconsider the content of this review and include some (or all if the number is high) the examples of 4f/5f-carbone/C metal complexes. This is because lanthanide organometallic chemistry has recently attracted the interest of several groups working in the areas of synthesis and molecule-based magnetism, with some Dy-C bonded complexes showing spectacular quantum properties.

- There is no discussion of the use of any of these complexes for any kind of application. I think that the authors could selectively discuss the catalytic (or any other) property of 2-3 complexes from this review in a new paragraph/section of this manuscript.

- In many cases within the manuscript the representation of the unpaired electrons of C as a vertical line (|) is misleading and it looks like an iodide or “Cl” (chloride). I recommend the authors to change this representation to the usual (..) two-dot representation for the unpaired electrons.

Reviewer 2 Report

This review by Petz and Frenking at al. summarizes experimental and theoretical studies on the chemistry of carbone- and carbide-transition metal complexes. The bonding situations and electronic structures of the reported carbone- and carbide transition metal complexes are well explained and compared, and hence, this review will help readers to understand the background and overview of this topic. However, this manuscript still has a huge room for improvement as well as numerous errors, although it is said that “All authors have read and agreed to the published version of the manuscript”. Moreover, there are serious problem on English (grammar, typos and so on) as well as interpretation of a cited paper (see below). Due to these problems this referee stopped reading the manuscript, and recommends the authors to resubmit after careful amendment. The followings are the points that need a reconsideration or correction.

page 2, caption in Figure 2

This referee is not sure what “central fragments” indicate. Does it mean central carbon atom? This point should be clarified. In addition, although the compounds in Figure 2 have various D-C-D angles, where D is a donor, the Chem Draw structures seem to have essentially the same D-C-D angles.

page 3, the last paragraph

More explanation is necessary for “but a linear form of 1a is realized if crystallized from benzene [28]”. Does it mean the solid-state structure of 1a is highly dependent on the solvent used for recrystalization? This point needs more clarification. Furthermore, the citation numbers in the main text does not match with those in the Reference section. [28] should be [29]. The authors should pay much more attentions to the citation numbers throughout the manuscript, and amend them.

page 7, 2.2, the first sentence

The first sentence “Starting material for 2a is not the free carbone Ph2P-CH2-PPh2-C-PPh2-CH2-PPh2 which could not be prepared so far, but the dication [Ph2P-CH2-PPh2-CH2-PPh2-CH2-PPh2]2+ as reported by Peringer [55]” is COMPLETELY WRONG. Again, [55] should be [56]. According to ref. 56, starting material for 2a is not the dication, and more importantly, how can we determine the starting material 2a that has not been isolated so far.The authors should read each reference more carefully. At this point, this referee stopped reviewing the manuscript.

Round 2

Reviewer 1 Report

Although I am totally against aggressive arguments between authors and reviewers (in this case between reviewer 2 and the corresponding author), I (as reviewer 1) believe that the authors have revised their ms according to most of my comments and I recommend the acceptance of this version for publication without any further revisions.